# Using Genomic Variation to Distinguish Ovarian High-Grade Serous Carcinoma from Benign Fallopian Tubes

**DOI:** 10.3390/ijms232314814

**Published:** 2022-11-26

**Authors:** Jesus Gonzalez-Bosquet, Nicholas D. Cardillo, Henry D. Reyes, Brian J. Smith, Kimberly K. Leslie, David P. Bender, Michael J. Goodheart, Eric J. Devor

**Affiliations:** 1Department of Obstetrics and Gynecology, University of Iowa, 200 Hawkins Dr., Iowa City, IA 52242, USA; 2Hanjani Institute of Gynecologic Oncology, Thomas Jefferson University, Philadelphia, PA 19107, USA; 3Department of Obstetrics and Gynecology, University of Buffalo, Buffalo, NY 14203, USA; 4Department of Biostatistics, University of Iowa, 145 N Riverside Dr., Iowa City, IA 52242, USA; 5Division of Molecular Medicine, Departments of Internal Medicine and Obstetrics and Gynecology, The University of New Mexico Comprehensive Cancer Center, 915 Camino de Salud, CRF 117, Albuquerque, NM 87131, USA

**Keywords:** genetic variation, ovarian cancer, prediction model, whole exome sequencing, RNA sequencing

## Abstract

The preoperative diagnosis of pelvic masses has been elusive to date. Methods for characterization such as CA-125 have had limited specificity. We hypothesize that genomic variation can be used to create prediction models which accurately distinguish high grade serous ovarian cancer (HGSC) from benign tissue. Methods: In this retrospective, pilot study, we extracted DNA and RNA from HGSC specimens and from benign fallopian tubes. Then, we performed whole exome sequencing and RNA sequencing, and identified single nucleotide variants (SNV), copy number variants (CNV) and structural variants (SV). We used these variants to create prediction models to distinguish cancer from benign tissue. The models were then validated in independent datasets and with a machine learning platform. Results: The prediction model with SNV had an AUC of 1.00 (95% CI 1.00–1.00). The models with CNV and SV had AUC of 0.87 and 0.73, respectively. Validated models also had excellent performances. Conclusions: Genomic variation of HGSC can be used to create prediction models which accurately discriminate cancer from benign tissue. Further refining of these models (early-stage samples, other tumor types) has the potential to lead to detection of ovarian cancer in blood with cell free DNA, even in early stage.

## 1. Introduction

The main reason why ovarian cancer remains one of the most deadly cancers in women in the United States [1] is because it is diagnosed in an advanced stage in over 70% of patients, with 5-year overall survival around 40–50% [2]. Patients diagnosed at early stages have significantly improved 5-year survival exceeding 90%. High grade serous carcinoma (HGSC) is the most common histologic type of the ovarian cancer spectrum, which includes fallopian tube, ovarian and primary peritoneal cancers. It is clear that early diagnosis of ovarian cancer would save lives. Unfortunately, no current method of screening has been proven effective at detecting ovarian cancer at an earlier stage [3].

Patients with ovarian cancer typically present with a pelvic mass, with or without symptoms or evidence of metastasis. In those without evidence of metastasis, it is difficult to determine if the mass is benign or malignant preoperatively. A biopsy of the mass is not advised as it has the potential to upstage the patient and may not obtain adequate tissue for diagnosis. Numerous medical organizations and groups have published recommendations on how to manage these pelvic masses [4,5,6,7]. Usually, clinical data, tumor markers and imaging studies (generally ultrasound) are used. Regrettably, the specificity of these algorithms varies depending on the utilized methods, and a large proportion of these masses are removed surgically, even when found by chance in asymptomatic women. The current standard to treat any suspicious masses is to surgically remove them, leading to surgery on a large number of benign masses which could have been safely observed, an expensive strategy to reduce ovarian cancer mortality. Furthermore, over 10% of these patients undergoing surgery will experience a major complication [8]. Ovarian cancer patients treated by trained gynecologic oncologists have improved outcomes over patients treated by other practitioners [9,10,11,12]. Unfortunately, it is estimated that one third of patients with gynecological malignancies never see a gynecologic oncologist [13]. Therefore, it is critical to better select patients with a high probability of ovarian cancer to be referred to trained specialists.

Cancers undergo a myriad of mutations and alterations in their genome during their development. While these mutations may not cause direct transcriptional or translational consequences, it is possible to use them to create models that would discriminate cancerous tissue from benign tissue. To achieve this goal, we leveraged a well annotated biobank of HGSC and normal fallopian specimens and determined and compared genomic variation of these samples. The goal of our pilot study was not to perform a comprehensive analysis of genomic variation, but to assess the feasibility of these variants, no matter how rare [14,15], to discriminate HGSC from normal tissue. Then, we validated our prediction models of HGSC with different platforms, independent datasets, and machine learning algorithms.

## 2. Results

### 2.1. Single Nucleotide Variation

The workflow for the SNV analysis is represented in Figure 1. SNVs were identified with the VEP and superFreq methods from DNA of 20 HGSC (cases) and 14 normal fallopian tube (controls).

#### 2.1.1. VEP Analysis

30,951,598 variants for all samples, located in 11,342,380 unique loci, were identified in our initial analysis in both the tumor and control specimens. A visual representation of all SNVs identified from just one sample and the downstream effects of these SNVs are shown in Appendix A. 242,320 loci were also identified in the gnomAD database (Figure 1) and were removed from the analysis, leaving 11,100,060 unique loci for model construction. Appendix A shows all SNVs detected in the gnomAD database with VEP. Multiple univariate analysis of SNVs located in these 11 million loci and comparing tumor versus control samples resulted in 16,631 SNVs associated with HGSC that were selected for the multivariate analysis (*p* < 0.001) (Figure 2A).

#### 2.1.2. superFreq Analysis

220,293. SNVs within 118,839 unique loci were identified in our specimens. 100,164 of these unique loci were also present in either gnomAD database or the VEP analysis. This left 18,675 unique loci identified in the superFreq database which were not identified in the VEP analysis (Figure 1). When comparing cases and controls for SNVs located in these loci, 5 were found to be below the cut-off level in multiple univariate analyses (*p* < 0.05) (Figure 2B). None of these 5 SNVs were informative in the final multivariate model for HGSC prediction and were discarded, though.

The pathway analysis from the KEGG database was performed with 7151 genes harboring all 16,636 SNVs identified in both analyses. Significant pathways contain numerous intracellular signaling systems and are depicted in Appendix A and Appendix A.

#### 2.1.3. Prediction Modeling

A multivariate lasso regression using selected SNVs from all univariate analyses, resulted in a model with 49 SNV, with a performance measured in AUC of 1.0 (Figure 3). One of these 49 loci conferred significant protection for ovarian cancer (OR = 0.77 or protection of 27%), while the risk for HGSC of all others was practically null (OR = 1.00).

#### 2.1.4. Validation in RNA-seq

RNA was extracted and sequenced from 112 patients with HGSC (cases) and 12 fallopian tube specimens (controls). In the VEP analysis of the RNA-seq data, we identified 105,296,475 variants for all samples, located within 3,276,351 unique loci. When compared to the 11,100,060 unique loci identified in the WES data, 1,254,958 unique loci were common to both the WES and RNA-seq datasets (Appendix A). Of the 16,631 selected SNVs associated with HGSC from the WES analysis, 6296 of these were also identified in the RNA-seq data and therefore common to both datasets. Using these common selected SNVs in the WES set identified in the RNA-seq analysis, multiple univariate regressions were performed to determine their association with HGSC. 532 were found to be associated with HGSC (*p* < 0.05) (Figure 4A). In the multivariate regression analysis, including these 532 SNV, three of these SNVs were independently associated with HGSC, all of which were protective (Figure 4B). A multivariate lasso regression model to predict HGSC was then created with all 532 SNV selected in all univariate analyses, resulting in a model with four SNV with an AUC of 0.99 (95% CI: 0.97–1.00) (Figure 4C).

*superFreq* SNV analysis was also performed using the RNA-seq data. However, none of the SNVs that were present in both the WES and RNA-seq data, but not present in the *VEP* analysis, passed the cut-off after initial univariate logistic analyses (Appendix A).

Pathway enrichment analysis was performed with genes harboring the 532 significant SNVs (Appendix A). The most significant pathways included the Fox0 pathway and GnRH signaling pathway (Appendix A).

Additionally, we used the TCGA database to further validate our results. When cross-referencing the unique loci in TCGA with the 16,631 identified in the VEP analysis with WES data, 8427 were common between the two datasets (Appendix A). No further comparisons were possible because of the lack of RNA-seq from controls in the HGSC TCGA set.

Validation of both models performed with machine learning (*TensorFlow*) classification with all significant SNV associated with HGSC (N = 16,631) and with SNV from the multivariate lasso model (N = 49) had an excellent performance, with an AUC of 1.00 (Appendix A). These models performed very well without any correction. Models with RNA-seq data also performed well but lower than those with DNA SNVs (Appendix A). The TensorFlow model with UI RNA-seq data, with 20 SNVs present in both DNA and RNA, had and AUC of 0.95, after accounting for unbalanced data (tubes represented less than 10% of the total samples). TCGA RNA-seq model, with 18 SNV present (out of the initial 49 SNVs from DNA) had an AUC of 0.93, also after accounting for unbalanced data (we used UI controls to construct the model).

### 2.2. Copy Number Variation (CNV)

CNV assessed using superFreq in the DNA WES identified 558 genes (out of the 23,443 genes assessed) that had different CNV between HGSC and controls (Figure 5A). The multivariate logistic regression determined that 2 of these CNV were independently associated with HGSC (Figure 5B). A multivariate lasso regression including all 558 significant CNV, resulted in a prediction model with 11 CNV that would predict HGSC with a performance by AUC of 0.87 (95% CI: 0.72*–*1.00; Figure 5C). Pathway enrichment analysis including all significant CNVs yielded one significant pathway, Herpes Simplex Virus 1 infection.

Validation of CNV results using RNA-seq data from UI specimens also was processed with superFreq. The 558 transcripts identified in the WES analysis were assessed (Appendix A). The validation of the CNV prediction model for HGSC had an AUC of 0.67 when using RNA-seq data (Appendix A). A lasso prediction model, independent of the DNA model, and constructed with the 588 significant CNVs with the RNA-seq data resulted in a prediction model of HGSC with two CNVs and an AUC of 1.00 (95% CI: 1.00–1.00). One gene significant in the multivariate analysis increased the risk for cancer and the other decreased the risk for HGSC (Appendix A).

In TCGA CGH database, 162 of the 588 CNVs significantly different between cases and controls in the WES analysis were present (Appendix A). 148 of these 162 were found to be significantly associated with HGSC as compared to normal controls. When a new lasso regression analysis was done in the WES CNV dataset using only CNV that were present in TCGA (N = 162) the resulting prediction model with 6 CNV had an inferior performance, with an AUC of 0.8 (95% CI: 0.74–0.86). When this model was validated in TCGA data the performance of the model had an AUC of 50% (Appendix A).

### 2.3. Structural Variation (SV)

In total, there were 250,886 SV of all types detected in the UI RNA-seq when comparing HGSC with normal tubes samples (Figure 6A). When comparing SV counts for each sample versus all controls, 32,156 were retained by MINTIE as significant SVs, at FDR < 0.05 and at least log2 fold (Figure 6B). Multiple univariate logistic regressions were performed to select 6003 of these SV associated with HGSC for the multivariate model (Figure 6C). 75 of these SV were decreased in HGSC and 5928 were increased (Appendix A). Multivariate analyses for all structural variations collectively as well as for each type of structural variation are reported in Appendix A.

A multivariate lasso regression analysis with all structural variations collectively identified 17 SV model that predicted HGSC with an AUC of 0.73 (95% CI: 0.69–0.77) (Appendix A).

TCGA validation detected a total of 101,567 SVs in HGSC RNA-seq samples versus all controls. Out of the 6003 SVs significantly associated with HGSC in the UI analysis, 3429 were present in TCGA set, and 3353 of these were associated with HGSC when comparing cases versus controls (*p* < 10^−5^) (Appendix A). A multivariate logistic regression analysis of TCGA significant SVs showed 2 SV independently associated with HGSC. A new lasso regression analysis was performed with UI data using only those 3429 SVs present in TCGA. It resulted in a prediction model with an AUC of 0.71 (95% CI: 0.63–0.79). Validation of this model with TCGA data performed very similar to the UI model with an AUC of 0.74 (95% CI: 0.71–0.77) (Appendix A).

## 3. Discussion

The overarching goal of this pilot study was to identify genetic/genomic variation that could differentiate HGSC from normal tubal tissue. With that aim, we used DNA and RNA sequencing and identified a comprehensive array of genetic/genomic variation in HGSC and benign fallopian tubes. Then, we used those DNA and RNA characteristics to create prediction models that would discriminate accurately and robustly, by validating those models in different settings, in independent databases, in different platforms, and with different analytics. Our SNV model carried the highest accuracy with an AUC of 1.00 when using 49 different SNV loci, while the models for copy number variation and structural variation were successful, but somewhat less accurate with AUC of 0.87 and 0.73, respectively. These models were then successfully validated using RNA-seq data from our dataset as well as TCGA, and the SNV data was even further validated using machine learning with TensorFlow. Initially, the SNV model is the most promising, but those created for CNV and SV were not without merit.

The SNV model contained 49 separate loci and had consistently excellent results in all methods of validation. Some of these variants were associated with a worse prognosis and some were associated with protection from ovarian cancer, but others have no known effects. The variants which offered protection from ovarian cancer were more frequent in the normal samples as compared to the HGSC specimens. This is proof that using infrequent variants in these specimens and coalescing them into a prediction model is a viable method for cancer detection, even if these variants are not known to have downstream consequences.

Ovarian cancer, and its most common type HGSC, are commonly diagnosed in advanced stage (75–80%). We built prediction models that would discriminate the most frequent presenting form of ovarian cancer. The resulting best model also offers the simplest method for reproducibility by others. For these models of HGSC prediction to be applied clinically, still several steps must be taken. First, they will have to be validated in prospective manner in a larger sample size to account for potential overfitting that could occur in retrospective analyses. Should this be successful, then this method could potentially be used to identify these variants in peripheral blood specimens from circulating tumor DNA (ctDNA) as a method for ovarian cancer diagnosis. Modifications of these models would have to be done to account for earlier stages of HGSC and for other rare forms of ovarian cancer. Adding earlier stages may result in a tool to diagnose all stages of HGSC. Including other types of ovarian cancer (germ cell, stromal cell types) may result in a broader detection tool for ovarian cancer. We believe that our study is the first step in that direction.

It is well known that circulating tumor DNA (ctDNA) and RNA can be isolated from peripheral blood draws in patients. Circulating tumor DNA is identifiable in ovarian cancer patient plasma with reliable accuracy, and SNV, CV, and SV have all been identified in peripheral specimens [16,17,18,19]. Conceivably, it is feasible to identify the variations used in the models through a peripheral blood draw on patients with pelvic masses of unknown etiology. ctDNA could then be amplified and the necessary variants identified using a pre-defined panel. ctDNA is a better medium for this type of testing because it is more stable in an extracellular form than RNA, though circulating tumor RNA has also been identified and sequenced. This constitutes a method to diagnose ovarian cancer with a “liquid biopsy”. ctDNA has been identified in ovarian cancer patients at early stages [20]. Panels for other cancers have been created to drive individualized treatment, cancer surveillance, and early diagnosis; two of which have recently gained FDA approval [21,22]. Given that ovarian cancer ctDNA is identifiable in patients with early-stage ovarian cancer, and that our model predicts ovarian cancer with high accuracy, the model described here has the potential to provide a strong diagnostic tool for ovarian cancer at an early-stage.

Additionally, this method could be used to identify recurrent or persistent cancer in patients who have completed their adjuvant therapy. Liquid biopsy has already been shown to be feasible in identifying recurrence in ovarian cancer and these methods are currently being investigated in the monitoring of lung cancer [23,24]. While these would likely require different models to identify those variations which are present in recurrent cancer, the methods for creating them would be the same.

The current methods for pre-operative characterization of pelvic masses leave significant room for improvement. A method with little to no false negative or positives would allow us to triage referrals to gynecologic oncologists appropriately, maximizing their efforts while keeping benign patients with their trusted gynecologists to either proceed with surgery for symptomatic masses or avoid surgery altogether for asymptomatic masses. This would lower the costs for surgery, referrals to specialists, and minimize risk to patients with benign masses.

The strengths of this study are a thorough analysis of all possible SNVs, CNVs and SVs in well annotated patients with HGSC and an adequate number of appropriate controls. These results were validated in independent datasets (TCGA), with different platforms (CGH, RNA-seq), and different analytical platforms maintaining acceptable performances. By analyzing different types of genomic variations, we were able to increase the chances that we could create an accurate prediction model. Some of the limitations of our study is that the specimens used were from advanced HGSC and benign fallopian tubes. While this allowed us to obtain a greater number of specimens and compare them to truly benign tissue, it limits our ability to apply these results to early-stage disease and other types of ovarian cancer. Continued efforts collecting more specimens of varied stages, histologies, increasing the racial diversity of our specimens, and adding more rare variants, will further improve the accuracy and generalizability of the prediction model.

## 4. Materials and Methods

We performed a single institution, retrospective, pilot study using tumor specimens obtained at the time of cytoreductive surgery from 112 patients with HGSC and compared them to benign fallopian tube specimens from 14 patients collected at the time of surgery for benign indications. DNA and RNA were isolated from all specimens and whole exome sequencing (WES) and RNA sequencing (RNA-seq) were performed. Using this WES and RNA-seq data, single nucleotide variants (SNV), copy number variation (CNV) and structural variation (SV) were identified. As mentioned before, the goal was not to perform a comprehensive analysis of genomic variation, but to assess the feasibility of these variants to discriminate HGSC from normal tissue.

Genetic variations, including somatic and germline mutations, were identified initially from WES performed in DNA extracted from 20 HGSC specimens and 14 normal Fallopian tubes specimens. Differences between variants from HGSC and tubes were assessed. Since tumor tissue was compared to non-matched control tissue, germline variants could not be identified separately from somatic variations. Then, models to predict HGSC were constructed using multivariate lasso regression analysis with variants that were different between HGSC and tubes. These models were then validated with RNA-seq data from all 112 HGSC cases and 12 of the controls (tubes), and with RNA-seq from The Cancer Genome Atlas (TCGA) database (all HGSC, N = 376). Prediction models were further validated using machine learning algorithms with *TensorFlow*.

### 4.1. Specimen Acquisition

HGSC tissue samples and clinical outcome data were obtained from the Department of Obstetrics and Gynecology Gynecologic Oncology Biobank (IRB, ID#200209010), which is part of the Women’s Health Tissue Repository (WHTR, IRB, ID#201804817). All specimens archived in the Gynecologic Oncology Biobank (herein termed Biobank) were originally obtained from adult patients under written, informed consent in accordance with University of Iowa (UI) IRB guidelines. Tumor samples were collected, reviewed by a board-certified pathologist, flash-frozen, and then the diagnosis was confirmed in paraffin at the time of initial surgery. All experimental protocols were approved by the University of Iowa (UI) Biomedical IRB-01.

We then collected fallopian tube samples from women undergoing gynecologic procedures. Fallopian tubes were obtained from patients with no family history of cancer beside squamous cell carcinoma of the skin and who were undergoing salpingectomy for benign indications (mainly sterilization). Fallopian tubes were chosen as controls as this is the most likely origin of HGSC [25]. DNA and RNA were extracted from epithelial tissue coming from the junction of the ampullary and fimbriated end of fallopian tubes. Twenty normal fallopian tube specimens were obtained. Of those, 12 produced viable RNA for analysis. RNA from both the fallopian tube and HGSC specimens had already been extracted and purified in a previous study [26]. WES was performed on 14 fallopian tube specimens.

483 patients with serous ovarian cancer were identified in the database. Only patients with a primary diagnosis of advanced Stage (III and IV) HGSC were included. Of these, 112 patients with fresh frozen specimens were selected for RNA isolation and genomic analysis, as described previously [27]. 20 patients with specimens with high quality DNA were also selected for WES to perform variation analysis.

### 4.2. DNA Sequencing

Genomic DNAs (gDNAs) were purified from frozen tumor and fallopian tube tissues using the DNeasy Blood and Tissue Kit according to manufacturer’s (QIAGEN) recommendations. Yield and purity were assessed on a NanoDrop Model 2000 spectrophotometer and by horizontal agarose gel electrophoresis. Whole exome sequencing (WES) was performed externally by GeneWiz (Azenta, Chelmsford, MA, USA) with 100× coverage. Mean quality score was 37.76 and the percent of bases greater than or equal to 30 was 89.96.

### 4.3. RNA Sequencing

RNA was isolated from the tumor and control specimens in the Biobank. RNA extraction, processing and sequencing have been described previously [27,28]. In brief, total cellular RNA was extracted from primary tumor tissue using the mirVana (Thermo Fisher, Waltham, MA, USA) RNA purification kit. The RNA yield and quality were assessed with Trinean Dropsense 16 spectrophotometer and Agilent Model 2100 bioanalyzer. RNA quality was determined to be adequate if the sample had an RNA integrity number (RIN) of 7.0 or greater. Samples that were of adequate quality were then sequenced. 500 ng of RNA was quantified by Qubit measurement (Thermo Fisher). RNA was then converted to cDNA and ligated to sequencing adaptors with Illumina TriSeq stranded total RNA library preparation (Illumina, San Diego, CA, USA). cDNA samples were then sequenced with the Illumina HiSeq 4000 genome sequencing platform using 150 bp paired-end SBS chemistry. All sequencing was performed at the Genome Facility at the University of Iowa Institute of Human Genetics (IIHG).

### 4.4. Single Nucleotide Variation (SNV) Analysis

DNA from WES was aligned to the human reference genome (version hg38) using Subread. BAM files resulting from the alignment were used with samtools [29] and VarScan software [30] to create Variant Call Format (VCF) files for further analysis, as recommended by best practices of genome sequencing [31]. Two separate methods were used to identify all possible SNV’s present in VCF files. The first method used was the Ensembl Variant Effect Predictor (VEP). This method determines if variants cause significant downstream changes in the genome [32]. Variants present upstream of transcripts, coding regions, regulatory regions, non-coding RNA, and that have downstream consequences (i.e.*,* missense, frameshift, stop gained or lost) are retained and inconsequential variants are removed. VEP is able of identifying rapidly 4*–*5 million variants (all types, including SNPs) present in a typical sequenced genome [32,33]. A table with present SNVs for all samples was constructed and used for analyses.

In parallel, VCF files were processed with *superFreq* package to detect other novel SNVs [34]. This package uses a series of filters to discard SNV with lesser quality. Additionally, SNV detected by *superFreq* that were present in the gnomAD database were removed. *SuperFreq* assessed SNV in each transcript of each sample and performed a log-ratio of transcripts with versus without SNV within that sample. That logRatio was used later for further comparative analyses.

All unique loci containing those SNV’s by both methods were codified.

To identify those SNV unique in HGSC we subtracted all common SNV present in a large, publicly available database, gnomAD, from SNV results from both VEP and superFreq (Figure 1). The gnomAD database (gnomad.broadinstitute.org) is a database containing 125,748 WES from various studies, aggregated by researchers worldwide [35]. Multiple univariate logistic regression analyses with each of the resulting SNVs were performed to identify those variants more informative for cancer (HGSC, a dichotomous dependent variable, y). The goal was to reduce the number of variables to be introduced in the multivariate prediction analysis to build a classifier, as recommended [36]. Independent variables (x) were the presence (or absence) of a particular SNV. The cut-off level was established at a *p*-value < 0.001. Using the variants identified in these multiple univariate analyses, an enrichment pathway analysis was performed using the clusterProfiler R package [37], which interrogates the KEGG database (https://www.genome.jp/kegg/pathway.html, accessed on 31 December 2021). Significance at the *p*-value level of < 0.05 were corrected for multiple comparisons using false discovery rate (FDR). Variants selected with these univariate analyses were included in a multivariate lasso regression analysis to create prediction models of HGSC. Performance of all prediction models were measured by the area under the curve (AUC).

Validation was performed in two different datasets: with RNA-seq data from UI specimens and with RNA-seq data downloaded from The Cancer Genome Atlas (TCGA). UI RNA-seq data was also aligned to the human reference genome (version hg38) with the STAR suite. As before, BAM files were used to create VCF files with samtools and VarScan, and SNVs were detected with both VEP and superFreq. SNVs from RNA-seq data were compared to those identified in DNA through WES. Only SNVs from the RNA-seq experiments that were present and selected in WES multiple univariate logistic analyses of HGSC versus normal tubes were used for validation analysis. These common SNVs were assessed for their association with ovarian cancer (HGSC) in multiple univariate logistic models (cut-off, *p*-value < 0.05). As before, selected SNVs in univariate analyses were used to create a prediction model of HGSC in the multivariate lasso regression analysis (Appendix A). In addition, pathway enrichment analysis using the KEGG database was performed with these selected SNVs in the univariate analysis.

After permission was granted to access controlled data by the Genomic Data Commons (GDC) Data Portal (dbGaP# 29868), TCGA HGSC BAM files from RNA-seq experiments aligned to the human reference genome (version hg38) with the STAR suite were downloaded in their original format. TCGA contains RNA-seq data but lacks RNA-seq from normal tubal samples. We were able, though, to create VCF files from original TCGA BAM files. SNVs were identified by performing VEP and superFreq analyses, in the same way as the WES analysis. Then, these SNVs were compared to those present and associated with HGSC in the WES analysis (Appendix A).

To further validate the analysis, we performed the prediction analysis in a machine learning platform. For that purpose, we used TensorFlow [38] in a Jupyter notebook with a Keras application programming interface (API) [39]. TensorFlow code was modified from a tutorial (found here: https://www.tensorflow.org/tutorials, accessed on 2 February 2022). Several models were tested in this platform: first, the model with all SNV associated with HGSC, the model resulting from the multivariate analysis with lasso, the model for the validation with IU RNA-seq SNVs, and the model for the validation with RNA-seq extracted from TCGA. We used classification machine learning approaches to discriminate tube versus HGSC. Training and testing were performed to account for weights of the outcomes as well as for unbalanced data (fewer controls than cases for both DNA and RNA samples).

### 4.5. Copy Number Variant (CNV) Analysis

Resulting DNA WES files from the Subread alignment (BAM) were assessed for copy number variations using superFreq [34]. Once CNV were detected, a univariate logistic regression was carried out to identify those CNV more informative for HGSC, with a cut-off *p*-value of <0.001. Enrichment pathway analysis was performed with genes that had significant CNV differences in the univariate analysis using clusterProfiler and the KEGG database. Significance was at a FDR < 0.05. Then, two multivariate analyses were performed with the significant CNV: (1) multivariate regression additive model to identify CNV independently associated with HGSC; (2) multivariate lasso regression analysis to identify which CNV predicted HGSC.

Validation was performed in two independent RNA-seq experiments. First, we identified CNV in the RNA-seq database from the UI with superFreq. The CNVs found to be significant in the univariate analysis of the DNA WES were assessed in the resulting RNA-seq CNV data. A lasso prediction model was created with these CNV resulting from the RNA-seq data, to validate the prediction model created with the significant CNV in the WES dataset. Additionally, a new multivariate lasso prediction model was created with the RNA-seq data, independent of the initial WES model, and their respective performances were compared. Finally, a multivariate logistic regression of the significant CNV from the WES data was performed in the RNA-seq data, to identify independently significant CNV.

The second validation was performed in TCGA HGSC database. Level 3, or segmentation files, resulting from TCGA HGSC dataset initial array comparative genomic hybridization experiments (CGH) were downloaded, as detailed previously [40]. This database was chosen because it has matched normal samples for the comparison. Circular binary segmentation was used to identify regions with altered copy number in each chromosome [41]. The copy number at a particular genomic location was computed based on the segmentation mean log-ratio data. The database was first assessed for the presence of significant CNV in the original, WES univariate analyses. With the resulting CNV new multiple univariate analyses were performed to determine which had significant changes in their copy number (*p*-value < 0.05), represented by their mean log-ratio. Using the genes with CNV present in the TCGA database, a new lasso prediction model was created with WES data, and then validated in this new TCGA data. Finally, a multivariate logistic analysis of all CNV present in TCGH CGH arrays and WES data was carried out to identify independently CNV associated with HGSC.

### 4.6. Structural Variation (SV) Analysis

RNA-seq data from UI database (both tumor specimens and controls) was used to assess structural variation using MINTIE [42]. RNA-seq data was used because rearrangements can be more reliably identified in the transcriptome, especially fusion transcripts [43]. MINTIE is an integrated pipeline for RNA-seq data that takes a reference-free approach, combining de novo assembly of transcripts with differential expression analysis to identify up-regulated novel variants in a case–control setting. Counts for each SV in each sample are compared with counts in all controls with the *edgeR* package. Then, significant SVs, with a false discovery rate (FDR) < 0.05 and log2 fold change >2 are retained, after which the transcripts are aligned to the genome and used for further analyses. Finally, SV counts are normalized, log transformed and used for univariate logistic regressions. SVs with differential expression (*p*-value < 0.001) were then selected and introduced in multivariate analyses, in an additive model first, to identify independently significant SV for HGSC, and in a lasso regression model later to identify predictors of HGSC. Significant variations were classified into type: fusion genes, alternative splicing, deletions, extended exons, intragenic rearrangement, insertion, novel exon, retained intron, novel exon, novel exon junction, and partial novel junction.

Validation was then performed in TCGA RNA-seq data and using UI RNA-seq from UI fallopian tubes for controls, because MINTIE does not need to run on matched normal tissues or controls. Initially, BAM files were converted to fastq files with bedtools for the analysis [44]. Then, MINTIE was used to identify all SVs in TCGA dataset. SVs significant in the UI dataset that were also present in TCGA were used for validation. Briefly, significant SVs in the comparison of normalized counts for each SV in each sample versus counts in all controls (corrected *p*-value with FDR < 10*^−^*^5^) were introduced in multivariate analyses, in an additive model first, to identify independently significant SV for HGSC, and in a lasso regression model later to validate UI predictors of HGSC. For this validation, we selected only those SVs that were present in both UI and TCGA databases: (a) first we constructed a prediction model of HGSC with lasso regression in the UI database with these common SVs; (b) then we validated the prediction model created in UI with TCGA data.

## 5. Conclusions

In this retrospective, feasibility pilot study, we identify a set of genomic variation (SNV) that can discriminate HGSC with high performance. This effort could be the first step to a detection tool for ovarian cancer in serum, potentially even in early stages. The final set of markers could be used to build a DNA genotyping chip, similar to other commercial DNA tests, where the patient to be tested could send a DNA sample by mail and would get her results in a couple of weeks.

## 6. Patents

A patent with the 49 SNVs resulting from the prediction model of HGSC is being completed at the time of this publication.

## Figures and Tables

**Figure 1 ijms-23-14814-f001:**
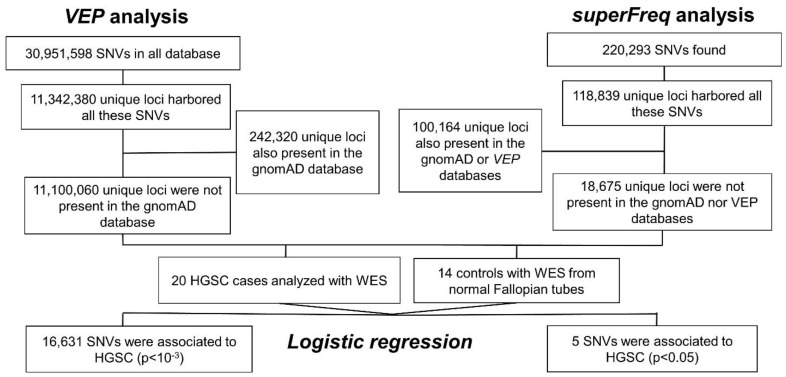
Variation analysis with Ensembl Variant Effect Predictor (VEP). Analysis was performed in DNA of 20 HGSC and 14 normal tubes. Of the initial approximately 31 million SNVs found in all HGSC samples, located in 11.3 million unique loci and the remaining were repeated, for an average of over 404 k SNVs for HGSC samples. Over 242,000 of them were already present in normal controls from the gnomAD database, with no differences in allele frequencies, *p*-value < 0.05, from more than 125,000 individuals. These were subtracted from the analysis. The resultant SNVs were assessed for their association with HGSC with multiple univariate logistic regressions. Unrelated controls consisted in 14 DNA sample from the distal part of the Fallopian tube (fimbria) from patients with no disease and no family history of ovarian cancer. The analysis resulted in 16,631 selected SNVs, associated with HGSC, at a *p*-value < 10^−3^. Variation analysis with superFreq. The analysis was performed in the same cases and controls than before. Of the initial approximately 220 thousand SNVs found in all HGSC samples after quality filters, 118,839 were in unique loci. Over 100,164 of them were already present in normal controls from the gnomAD database, or in the previous analysis with VEP. These were subtracted from the analysis. The resultant 18,675 SNVs were assessed for their association with HGSC with multiple univariate logistic regressions, as in VEP analysis. 5 significant SNVs, associated with HGSC were selected, at a *p*-value 0.05. SNV: single nucleotide variation; HGSC: high-grade serous ovarian cancer; WES: whole exome sequencing.

**Figure 2 ijms-23-14814-f002:**
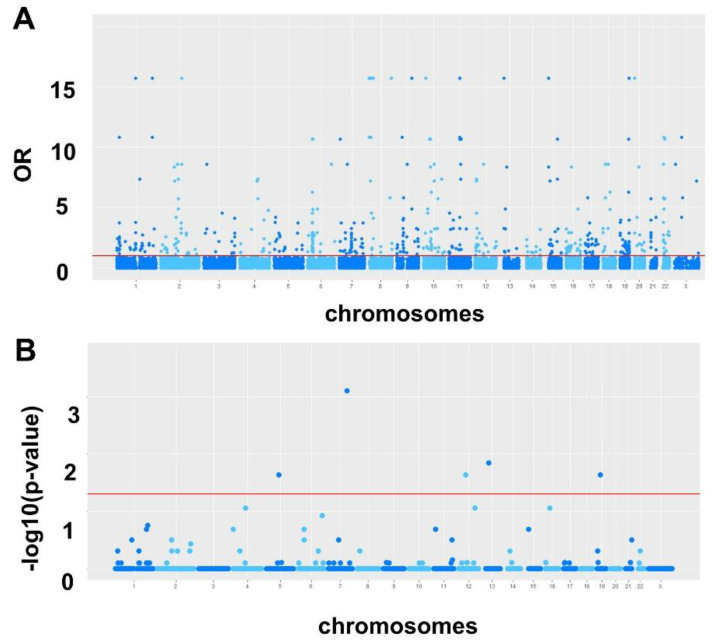
Variation analysis with VEP and superFreq. (**A**). Manhattan plot representation of the resulting 16,631 selected SNVs associated with HGSC from multiple univariate logistic regressions (selected cut-off at *p*-value < 10^−3^). The x axis represents location of SNV within chromosomes; the y axis represents the ORs (Odds Ratio) of the association with HGSC; the red line designates *p* = 10^−3^; (**B**). Representation of 5 selected SNVs (out of 18,675) associated with HGSC, at a *p*-value <0.05, after superFreq SNV determination and univariate logistic regression analyses comparing HGSC and normal tube. These SNVs were not identified within the VEP analysis. The x axis represent location of SNV within chromosomes; the y axis represents the log transformation of the *p*-value.

**Figure 3 ijms-23-14814-f003:**
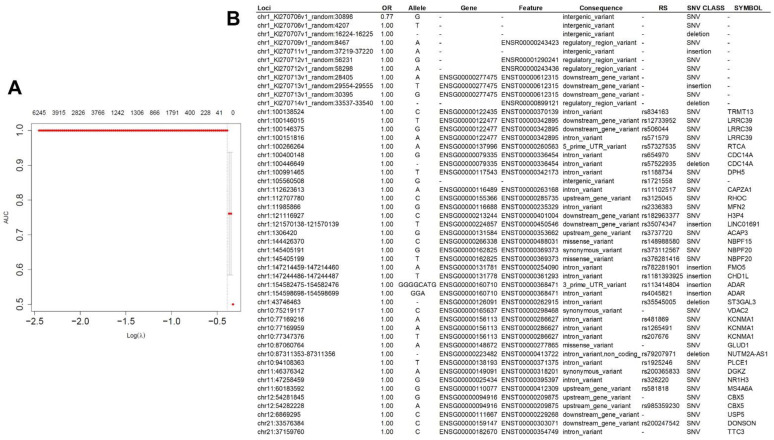
Lasso multivariate regression analysis of 16,636 selected SNVs in all univariate analyses. (**A**). Graphic representation of the lasso prediction model: superior margin reflects number of variables; left margin reflects performance of the model measured in AUC (area under the curve); lower margin reflects lambda (λ) tunning parameter chose by cross-validation to optimize the model. The model had a performance, measured in AUC of 1.0, when it included 49 SNVs. Details of these SNVs are represented in the table; (**B**). Only one locus conferred substantial protection to HGSC: chr1_KI270706v1_random:30898, (allele G). The other loci’s risks were insignificant. The majority of these SNVs have been already described (RS column). λ: lambda is a tunning parameter of the lasso regression (when λ = 0, the shrinkage penalty has no effect).

**Figure 4 ijms-23-14814-f004:**
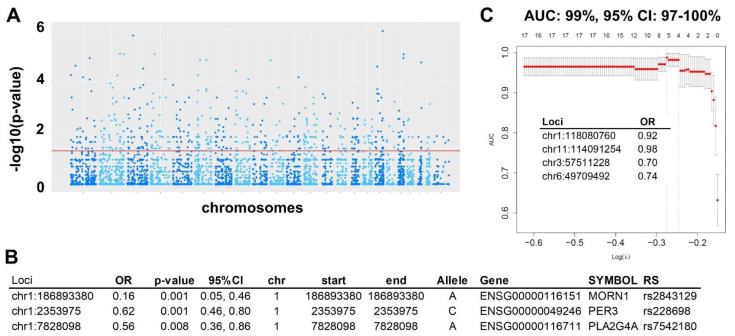
Variation analysis with VEP and superFreq in RNA-seq experiments. (**A**). 6296 SNVs were also present in RNA-seq VEP analysis (out of 16,631 selected SNVs, associated with HGSC). In multiple univariate analyses, 532 SNVs were associated with HGSC and selected for multivariate modelling (at *p*-value < 0.05). (**B**). Multivariate logistic regression with all 532 SNVs. Three loci were independently associated with HGSC (*p* < 0.05) in this multivariate analysis. These 3 loci represent 3 SNPs already described (column RS), and all of them conferred protection against HGSC. (**C**). Multivariate lasso regression prediction model independent of DNA model: We introduced all 532 SNVs selected from multiple univariate analyses in a multivariate lasso prediction model. The model selected 4 loci: chr1:118080760, chr11:114091254, chr3:57511228, and chr6:49709492 that predict HGSC with an AUC of 99%.

**Figure 5 ijms-23-14814-f005:**
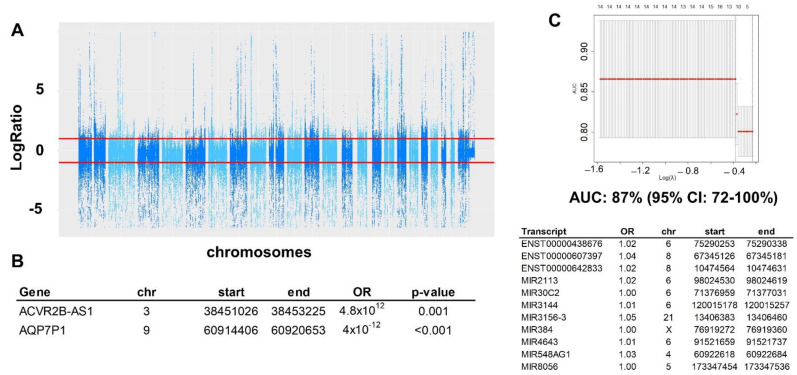
Gene copy number (CNV) analysis: there were 558 genes with differential copy number out of all 23,443 genes, at a *p*-value <10^−3^. (**A**). Manhattan plot representations of all 558 CNV. The red lines represent x2 copies (or more than diploid) and 0.5 copies (or less than heterozygous). (**B**). Multivariate logistic regression analysis to determine independently CNV associated with HGSC. (**C**) ROC curve of the multivariate prediction model with 11 transcripts with CNV: AUC of 87%, 95% CI:72%,100%). All of them increase the risk with relative moderate OR. Below the ROC is the table with transcript name and location.

**Figure 6 ijms-23-14814-f006:**
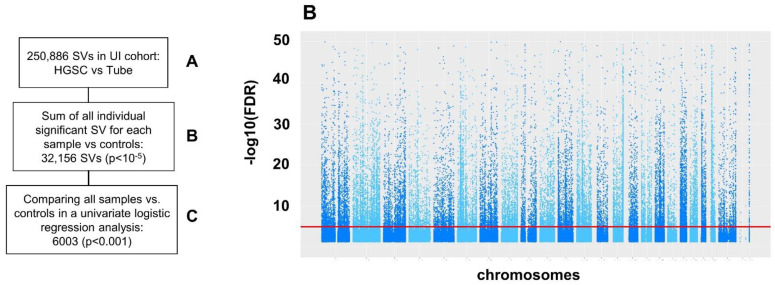
Comparison of structural variation (SV) between HGSC samples and tubal controls. Analysis performed with MINTIE: an integrated pipeline for RNA-seq data that takes a reference-free approach, combining de novo assembly of transcripts with differential expression analysis to identify up-regulated novel variants in a case–control setting. (**A**) All SV for each individual case (122), each one compared to controls (12). (**B**) Selected SV in the left panel are also represented in the right panel with a Manhattan plot. MINTIE selects those SV different between each case and controls, with a FDR <0.05 and log2 fold change >2. (**C**) Then, all selected SVs at the sample level were introduced in logistic models to compare all cases vs. all controls.

## Data Availability

Clinical data is not publicly available due to patient privacy. Datasets can be browsed by their accession number: GSE156699. The validation part of this study was performed in silico, with de-identified publicly available data. All data from TCGA is available at their web-site: https://portal.gdc.cancer.gov/, accessed 20 January 2022. Software utilized by this study is also publicly available at Bioconductor website: http://bioconductor.org/, accessed 20 January 2022.

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
