# Peer review of "Using Genomic Variation to Distinguish Ovarian High-Grade Serous Carcinoma from Benign Fallopian Tubes"

_ijms, 2022, doi:10.3390/ijms232314814_

Round 1

Reviewer 1 Report

Dear Authors,

Very interesting manuscript. The whole concept of research is logical and thoughtful.

I only have a reservation about the conclusion that should be expanded.

Wyniki tłumaczenia

Author Response

I only have a reservation about the conclusion that should be expanded.

We added an expanded vision of where this research is leading (in Conclusions section).

Reviewer 2 Report

In this manuscripts, authors tries to develop a predictive model of cancer diagnosis based on SNV, CNV and SV.  Overall this manuscript misses many key citations associated with the analysis and dataset, which makes the scientific evaluation and reproducibility very difficult.  In addition, readers will benefit with more explanation of each analysis.     

Major comments

Fig 1: What are the starting samples for VEP and superFreq?  Both methods says SNVs found in all HGSC samples with vastly different numbers.  From the text, it is not clear which data sets are used for each method especially if there are no associated citations.

Fig 2a/5a: I do not think p=0.001 is adequate for over 16k SNVs.  Simple first Bonferroni correction would be p=0.05/16000~3X10^-6. 

Fig 2b: Are these 5 SNVs statistically significant?  If this is based on the raw p-value, all of them will not be significant.  There are many SNVs in the analysis (n=18675?).  For multiple comparison, authors should use adjusted p value, q value, or Bonferroni correction.      

Lines 125-128 and Fig 3: Please, state what you are comparing (univariate vs multivariate).  What is the significant protection for cancer and its risk? 

Fig 4A: Again the results may be very different if multiple comparison is incorporated.

Minor comments

Fig 1: Please, define each abbreviations (e.g. WES).

Line 92-122: Please, cite all the references related with VEP, superFreq, and gnomAD.  Also, some details about each methods will be helpful for readers.

Line 228: Why is the cutoff p value 10^-5 here?

Author Response

Major comments

  1. Fig 1: What are the starting samples for VEP and superFreq? Both methods says SNVs found in all HGSC samples with vastly different numbers. From the text, it is not clear which data sets are used for each method especially if there are no associated citations.

We added the number of patients (20 HGSC cases and 14 normal tube controls) to Figure 1 capture and manuscript’s text, Results section.

  1. Fig 2a/5a: I do not think p=0.001 is adequate for over 16k SNVs. Simple first Bonferroni correction would be p=0.05/16000~3X10^-6.

We apologize for the confusion. We will make the process of feature selection for prediction modeling clearer across the manuscript. For Figure 2/5 specifically, first, we performed a univariate logistic regression analysis to identify those SNVs that were more informative for cancer and to reduce the number of variables to be introduced in the multivariate analysis. The p-value was supposed to be a selection cut-off for variables that would be introduced in the multivariate analysis, and not for a multiple comparison correction, i.e. feature selection. The analysis was performed for both SNVs resulting from the VEP and the superFreq analysis. Then, all significant/selected SNVs in the univariate analysis (16,631 for VEP and 5 for superFreq) were introduced in the multivariate lasso regression model to identify unique predictors of cancer (HGSC). Same was done for CNV and similar corrections were done to the Results and Methods sections. We added a new reference to add details about the selection method (Ref# 36: Simon, R., Roadmap for developing and validating therapeutically relevant genomic classifiers. J Clin Oncol 2005, 23, (29), 7332-41).

We reviewed the manuscript to clarify this point (Results and Methods).

  1. Fig 2b: Are these 5 SNVs statistically significant? If this is based on the raw p-value, all of them will not be significant. There are many SNVs in the analysis (n=18675?).  For multiple comparison, authors should use adjusted p value, q value, or Bonferroni correction. 

As before, the goal of this univariate analysis was to select features informative for HGSC. These 5 SNV were latter introduced in the multivariate analysis to build a prediction model for HGSC.     

  1. Lines 125-128 and Fig 3: Please, state what you are comparing (univariate vs multivariate). What is the significant protection for cancer and its risk?

Is a multivariate lasso regression prediction model, including selected features from the univariate analysis. We clarified this issue in the text. Protection and risks also were added to the manuscript.

  1. Fig 4A: Again the results may be very different if multiple comparison is incorporated.

We used the same approach that we used in DNA data for the RNA-seq data: first we selected the SNVs with a univariate analysis, and those that passed the cut-off were introduced in a multivariate regression analysis for prediction or association. We clarified this issues in Figure 4, and Results and Methods sections.

Minor comments

  1. Fig 1: Please, define each abbreviations (e.g. WES).

Done.

  1. Line 92-122: Please, cite all the references related with VEP, superFreq, and gnomAD. Also, some details about each methods will be helpful for readers.

The references are in the Methods section: REF# 32 and 33 for VEP; REF# 34 for superFreq; and REF# 35 as well as the website for gnomAD database.  Details about those methods are at the Methods section now.

  1. Line 228: Why is the cutoff p value 10^-5 here?

We apologize, it was our mistake, it was meant to be FDR < 0.05: MINTIE performs a comparison between each case and controls to determine counts of transcripts with structural variation (SV) using R package edgeR. Of all SV identified, MINTIE only retains those with FDR < 0.05 and log2 fold change >2. Those are the 32,156 SV described in Figure 6.

We explained better how MINTIE selects significant SV (in Methods) and how those selected SVs are used latter for our analysis. Also, we corrected the Results section and Figure 6.

Reviewer 3 Report

As ovarian cancer remains a real threat for women all studies contributing to change this situation are worth of attention. The reviewed study targeted into providing an analysis of genomic variation for a real distinguishing of high-grade serous carcinoma from benign fallopian tubes.  SNP was found as the best tool (prediction model has reached AUS of 1.00) as compared with CNV and SV. The authors suggest that it could be developed further into distinguishing particular steps of the disease.

In any case one has to remember that a limitation is an access to a molecular laboratory capable to analyze a necessary genomic variation. The author mentioned that one third of patients never see gynecological oncologist. So, how many samples would reach high level molecular lab? It would be worth to inform about time period necessary to analysis.

How many samples/patients were analyzed within your study as different numbers appear at page 10 (line 349 and below, line 356, line 381 and below).

References 35 and 39 seem to not have sufficient bibliographic information.

Author Response

  1. In any case one has to remember that a limitation is an access to a molecular laboratory capable to analyze a necessary genomic variation. The author mentioned that one third of patients never see gynecological oncologist. So, how many samples would reach high level molecular lab? It would be worth to inform about time period necessary to analysis.

This a great question, but we are not quite there yet. Once we have loci that have been validated as good classifiers of HGSC, we would do a specific DNA genotype chip with those 49 candidates loci. The patient to be tested could send a sample to a lab (spit, blood, ...), the DNA will be extracted and amplified, and then hybridized in the chip. This type of technology is available nowadays to mid-level laboratories and it is used commercially to determine ancestry, like 23andMe® or Ancestry®. The whole process would not take more than 2-3 weeks when is fully commercialized, even when the specimen is shipped to a lab.

We will add a few sentences in the Conclusions sections about where this research is leading and how do we see this genomic variation testing will play in the real world (complementary of Reviewer 1 comments).

  1. How many samples/patients were analyzed within your study as different numbers appear at page 10 (line 349 and below, line 356, line 381 and below).

We apologize for the confusion. We performed the analysis on 112 patients with HGSC and on tubal specimens from 14 women with no personal or familial history of cancer:

- DNA was extracted from 20 of the HGSC specimens and WES was performed on them. RNA was extracted from all 112 patients and RNA-seq was performed on them.

- DNA was extracted from 14 normal tube specimens and WES was performed on them. RNA was extracted on 12 tubal specimens and RNA-seq was performed on them. There was not enough RNA to do RNA-seq in the other 2 samples.

We clarified these issues in Methods and Results sections.

  1. References 35 and 39 seem to not have sufficient bibliographic information.

Corrected

Round 2

Reviewer 2 Report

The revision had improved the understanding on what authors have done.  Still I do not think the current writing and information provided is enough to validate the authors' findings. The manuscript is still poorly references, and it would be helpful to cite the source of information/data at the first incidence especially in the main text.  Authors tend to use several jargon directly from their analysis packages, but often a detailed explanation of input/output is lacking.

Major comments

1. associated with Figure 1. How is the logistic regression done?  I assume one axis is the frequency in HGSC, but the other axis seems unclear.  Authors will need a supplementary figure to demonstrate this (same for Fig S5). 

2. associated with Figure 2. I am still not convinced that "5" SNVs are truly significant here.  This is "5" out out almost 20k loci.  To me, this seems to suggest there is no association between these loci and HGSC.  By random chance, five loci is likely to be selected.  Authors need to provide additional evidences like a QQ plot. 

3. Lines 131-135: This part needs more explanation.  What is the input and output of this prediction? What are these 49 SNVs?   How does the AUC change as you add more loosely associated SNVs?

4. Lines 149-159: How do you test the HGSC association with RNA-seq?  A supplementary figure with an example would be helpful.

Minor comments

1. Figure 2: What is OR?

2. Lines 125-128: Is there an additional filtering for these 16,636 SNVs from N=18,675?

3. Figure 3: What is lambda?

4. Figure 5C: What is the x-axis?

5. Figure 6B: If FDR was used for selection, mark those points with different color rather than the red p-value cutoff.

Round 3

Reviewer 2 Report

The manuscript has improved.  The written explanation is still not very clear and can be confusing to readers.